# Role of Creatine in the Heart: Health and Disease

**DOI:** 10.3390/nu13041215

**Published:** 2021-04-07

**Authors:** Maurizio Balestrino

**Affiliations:** 1Dipartimento di Neuroscienze, Riabilitazione, Oftalmologia, Genetica e Scienze Materno-Infantili (DINOGMI), University of Genoa, Largo Daneo 3, 16132 Genova, Italy; mbalestrino@neurologia.unige.it; Tel.: +39-010-353-7040; 2IRCCS Ospedale Policlinico San Martino, Largo Rosanna Benzi 10, 16132 Genova, Italy

**Keywords:** phosphocreatine, creatine transporter, supplementation, treatment, heart, heart failure, ischemia, myocardial infarction, anthracycline, cardiac toxicity

## Abstract

Creatine is a key player in heart contraction and energy metabolism. Creatine supplementation (throughout the paper, only supplementation with creatine monohydrate will be reviewed, as this is by far the most used and best-known way of supplementing creatine) increases creatine content even in the normal heart, and it is generally safe. In heart failure, creatine and phosphocreatine decrease because of decreased expression of the creatine transporter, and because phosphocreatine degrades to prevent adenosine triphosphate (ATP) exhaustion. This causes decreased contractility reserve of the myocardium and correlates with left ventricular ejection fraction, and it is a predictor of mortality. Thus, there is a strong rationale to supplement with creatine the failing heart. Pending additional trials, creatine supplementation in heart failure may be useful given data showing its effectiveness (1) against specific parameters of heart failure, and (2) against the decrease in muscle strength and endurance of heart failure patients. In heart ischemia, the majority of trials used phosphocreatine, whose mechanism of action is mostly unrelated to changes in the ergogenic creatine-phosphocreatine system. Nevertheless, preliminary data with creatine supplementation are encouraging, and warrant additional studies. Prevention of cardiac toxicity of the chemotherapy compounds anthracyclines is a novel field where creatine supplementation may also be useful. Creatine effectiveness in this case may be because anthracyclines reduce expression of the creatine transporter, and because of the pleiotropic antioxidant properties of creatine. Moreover, creatine may also reduce concomitant muscle damage by anthracyclines.

## 1. Metabolism and Role of Creatine

### 1.1. Functions of Creatine

Creatine plays a key role in cellular energy metabolism. The creatine kinase enzyme reversibly phosphorylates it to phosphocreatine. Then, when phosphocreatine is reverted to creatine, its phosphate bond breaks, and such a break provides enough energy to allow phosphorylating a molecule of adenosine diphosphate (ADP) to adenosine triphosphate (ATP). Thus, phosphocreatine acts as an energy reserve to synthesize ATP rapidly, with no need for oxygen. The reaction is the following one:Creatine + ATP ⇆ Phosphocreatine + ADP + H^+^

This reaction plays a crucial role in heart contraction [1]. Its roles have been reviewed elsewhere [2] and are, in summary:

(1) Transfer of ATP from its production site (mitochondria) to its place of exploitation (neuronal membrane or cytoplasm). This function is often called “the ATP shuttle”. To carry out this transport (“shuttle”) of ATP energy, creatine first receives the phosphate from ATP near the mitochondria, becoming phosphocreatine. It then diffuses through its concentration gradient to the periphery of the cell. Near cytoplasmic ATPase, it donates its phosphate to ADP, effectively forming ATP far away from the mitochondria and delivering it precisely where and when it is required. In doing so, it reverts to creatine and diffuses back, again along its concentration gradient, to the mitochondrion to start the cycle again. The reason why the cell needs this complex mechanism to transport energy between mitochondria and cytoplasmic ATPase is that ATP is a very large molecule, therefore its diffusion through the organelle-filled cytoplasm is slow and cumbersome. By contrast, phosphocreatine is a much smaller molecule, thus it diffuses more quickly through the cytoplasm.

(2) Restoration by phosphocreatine of ATP concentration in conditions of increased energy demand and in diseases involving a reduced supply of blood or oxygen. In the first scenario, the consumption of ATP is excessive compared to the ability of the cell to synthesize it. For instance, a muscle exposed to a particularly intense effort quickly uses more ATP than it can produce, thus exhausting its reserve. In the second scenario, an organ cannot produce enough ATP because of a blood deficiency (ischemia) or an oxygen deficiency (anoxia). For example, in case of a myocardial infarction, phosphocreatine intervenes by transferring its phosphoric group to ADP, to provide ATP at a time when the heart cannot synthesize it due to ischemia.

Among all the biochemical reactions that our cells use to synthesize ATP, the one that starts from the creatine/phosphocreatine system is the quickest in buffering ATP levels at times of increased energy expenditure [3]. This explains the researchers’ interest in this molecule, whose administration has been proposed in various conditions, both physiological and pathological [4,5].

### 1.2. Procurement of Creatine by the Organism, with Specific Reference to the Heart

Creatine is normally degraded to creatinine [6], leading to a steady depletion of the body creatine store. The creatine store is replenished partly from endogenous synthesis and partly by ingesting creatine with food [7].

#### 1.2.1. Endogenous Synthesis of Creatine

In the body as a whole, creatine is synthesized in the kidney and in the liver [6]. Specifically, the kidney accomplishes the first step of the synthesis, forming guanidinoacetic acid from arginine and glycine. Guanidinoacetic acid is then transported to the liver, where it is converted into creatine with the intervention of the methyl donor S-adenosyl-methionine [6]. Specific organs can additionally synthesize creatine for their own consumption, such as the brain [8] and the testis [9,10]. Concerning the heart, it is generally believed that cardiomyocytes cannot synthesize creatine [11,12]. However, there is evidence that such synthesis may actually occur, i.e., cardiomyocytes may indeed synthesize creatine just as other organs do. In fact, the heart expresses the first enzyme of creatine synthesis, arginine-glycine amidino transferase (AGAT). Its expression in the heart is comparable to that of other tissues under basal conditions and increases several fold in pathological states [13]. Moreover, experiments showed that the addition of arginine (a precursor of creatine) to the incubation medium of toad hearts mitigated the decrease in creatine upon in vitro incubation, as if arginine was metabolized to creatine [14]. Furthermore, the addition of arginine to isolated rabbit hearts caused an increase in their content of creatinine (the product of creatine cyclization) [15]. Both these latter experiments strongly suggested that in the isolated heart arginine is indeed converted into creatine, as it is in other organs [16,17]. However, the possible creatine synthesis by the heart has received so far very little attention, and additional research is definitely warranted [18].

#### 1.2.2. Uptake of Creatine from Dietary or Supplement Sources

About half of the creatine that the organism needs is normally ingested and taken up from dietary sources [19]. Creatine is not present in vegetables, but it is only present in foods of animal origin [4]. Thus, subjects who do not regularly consume meat or fish tend to have some degree of creatine deficiency, and should supplement their diet with it [4].

Moreover, exogenous supplementation of creatine permits administering high amounts of this compound, and increasing its content above normal levels [20,21,22]. Throughout the paper, only supplementation with creatine monohydrate will be reviewed, as this is by far the most used and best-known way of supplementing creatine. When administered in this way at adequate doses, creatine is stored in the tissues, where it increases the intracellular pool of both creatine and phosphocreatine. Such an increase is especially relevant for the muscular tissue. Creatine supplementation allows the muscles to contract more powerfully and to a longer extent [23], an effect that is exploited by athletes to improve their performance [24,25].

The metabolism and functions of creatine in the heart is similar to that in the muscle and in other tissues. Figure 1 provides a summary of the metabolism of creatine in the myocardium.

Specific mechanisms of the benefit provided by creatine supplementation include:Restoration of normal creatine content when it is lower than normal due to lifestyle (e.g., vegetarian or vegan subjects [26]) or to disease (e.g., heart failure, see below).Increase in energy availability (obtained by increasing phosphocreatine concentration in the tissue) in cases where the balance between energy availability and requirement is limited by decreased energy production (as is the case in hypoxia or ischemia), or by increased demand (e.g., the muscle of athletes during athletic performance).

Finally, we should note that creatine by itself does not enter cells, instead it needs a specific transporter to cross plasma membranes [16]. The same happens in the heart, where the creatine transporter is present on the plasma membrane of the myocytes and is necessary for creatine to enter myocardial cells [27,28].

## 2. Cardiac Effects of Creatine Supplementation in Healthy Subjects

### 2.1. Cardiac Effects of Creatine Supplementation in the Normal Heart

#### 2.1.1. In Vitro Studies

Before discussing the effects of creatine supplementation in normal subjects, we shall mention some effects of creatine supplementation in in vitro normal heart preparations.

In heart strips prepared from frog ventricles, the force of contraction was increased when the preparation was perfused with a rather high concentration of creatine (9.2 mM) [29]. In an in vitro frog ventricular preparation, perfusion with rather high creatine concentrations (10 and 20 mM) reversibly caused, among others, increased force of contraction [30]. Interestingly, in both papers force of contraction was, by contrast, decreased when the in vitro preparation was perfused with much higher creatine concentrations (20–70 mM) [29,30], thus suggesting harmful effects by very high intracellular creatine increase. Santacruz et al. [31] reported increased content of phosphocreatine in in vitro cardiomyocytes upon creatine supplementation. Interestingly, these authors also reported such an increase in hypoxic conditions as well, although hypoxia decreased creatine uptake. Kilian et al. [32] used creatine supplementation in in vitro isolated hearts. They found that such a supplementation decreased heart rate, increased left ventricular systolic pressure, increased coronary flow, increased ATP content and decreased isocitrate dehydrogenase, a marker of cell death. Besides these positive findings, the authors add a puzzling statement (contained in the abstract and in the discussion sections, not in the methods nor in the results ones) saying that “when glucose supply was limited, conduction abnormalities occurred at a greater frequency in creatine-supplemented hearts as compared with the control group”. Surprisingly, there are no data presented to support this sentence, so the value of this statement is unclear. Summing up, these in vitro data suggest that supplementation with creatine of the normal heart may improve some physiologically relevant parameters, like force of contraction and coronary flow. However, they suggest that very high increase in creatine content may cause decrease in the force of contraction. We should emphasize that such harmful effects occurred only when the in vitro hearts were perfused with creatine concentrations extremely high, probably impossible to obtain in vivo (20–70 mM).

#### 2.1.2. In Vivo Studies

The normal heart has a high creatine content in comparison to other organs, as all excitable tissues have. For example, Horn et al. [33] reported in rats an average content of creatine about 80 nmol/mg protein in heart and brain, compared to about 20 nmol/mg protein in the liver and 25 nmol/mg protein in the kidney. Ipsiroglu et al. [21] found, in guinea pigs, a total creatine content (mean ± standard deviation) of 12.9 ± 0.10 µmol/g weight in the heart, 10.1 ± 0.45 µmol/g weight in brain, 7.5 ± 0.47 µmol/g weight in the liver and 5 ± 0.60 µmol/g weight in the kidney. Corresponding levels varied slightly in mouse and rat, but the proportions remained unchanged [21]. Human heart has an even higher creatine content. Neubauer et al. [34] confirmed a creatine content in the rat heart of 87.5 ± 4.2 nmol/mg protein, but found a content of 136.4 ± 6.1 nmol/mg protein in humans. The latter finding was confirmed by Nascimben et al. [35], who reported a creatine content of 131 ± 28 nmol/mg protein in human hearts. Such a high creatine content leads to downregulation of the creatine transporter and, it was stated, to little possibility of further increasing creatine content [12]. In agreement with this view, two rodent studies by a single laboratory concluded that creatine supplementation is unable to increase heart creatine content in the normal heart [33,36]. These studied concluded that a different strategy is needed to augment heart creatine, namely use of creatine derivatives that may enter cardiac myocytes with no need of the creatine transporter. It should be noted, however, that in neither of the two papers [33,36] creatine was increased in the muscles, a tissue where, by contrast, creatine is usually reported to increase upon adequate supplementation in both humans and animals [20,37,38]. Thus, the fact that in those two papers creatine did not increase not only in the heart but also in the muscle strongly suggests that the amount of creatine used in those studies [33,36] may have been insufficient. Accordingly, they reported creatine supplementation not as absolute daily amount (grams/day) but as percent of creatine in the total feed. Since the authors did not quantify the amount of food consumed daily, it is impossible to judge how large (or how insufficient) such supplementation might have been. By contrast, other authors reported increase in creatine content in normal hearts supplemented with creatine. Boehm et al. [39] found an 11% increased total creatine (i.e., creatine + phosphocreatine) in the hearts of animals whose feed was supplemented with 3% creatine. In this latter paper, ATP content was not affected by creatine supplementation, strongly suggesting that the creatine-phosphocreatine system created a reserve energy that might have possibly been used at times of increased energy demand or insufficient energy production. Ipsiroglu et al. [21] also found an increase in the creatine content of normal rodents’ heart upon creatine supplementation, and emphasized that such increase required a 4-week supplementation to become statistically significant. The increase reported by Ipsiroglu et al. [21] was, after 4 weeks of supplementation, 15% in the guinea pig, 17% in the mouse, 28% in the rat. Summing up, these studies suggest that increase in creatine upon its supplementation in the normal heart is possible, although one should administer high doses for several weeks, possibly because the normal heart already contains a large amount of creatine.

A different approach to experimentally increasing creatine in the normal heart involved genetically modified rodents where the creatine transporter was overexpressed, thus increasing intracellular creatine. The research group that first used this approach to increase myocardial creatine reported that increasing creatine transporter expression did cause increased intracellular creatine up to four times its basal value. However, they unexpectedly found that such increase in creatine content in normal hearts was attended by left ventricular hypertrophy and heart failure [40,41]. Those authors tentatively explained such apparent paradox with the fact that a very large increase in intracellular creatine required large amounts of ATP to phosphorylate creatine, and that this process caused a large decrease in intracellular ATP that, those authors argued, ended up by being detrimental to the heart [40,41]. Moreover, they found that such an increase in intracellular creatine caused downregulation of the cellular enzyme enolase, which supposedly decreased glycolytic activity of the cardiomyocytes [40]. However, a different group later used the same transporter over-expression to obtain a large increase in intracellular creatine, but did not find any adverse effect of this increase [42]. In an interesting study, Zervou et al. [43] compared wildtype mice, mice overexpressing the creatine transporter that showed a moderate increase in creatine, and mice overexpressing the transporter that showed a large creatine increase (average creatine content was 81 nmol/mg protein in wildtype animals, 123 and 220 in the other two groups, respectively). They found harmful effects only in the group with the maximal creatine increase, not in the group with the moderate creatine increase [43]. Furthermore, Lygate et al. [11] reported no adverse effect by overexpressing creatine transporter with an only moderate increase in creatine content. These data suggest that harmful effects on cardiac performance may unexpectedly accompany a very large creatine increase, but that this risk seems not to exist at all when creatine increases only by about 60% of its baseline level [43]. It is worth noting that in the human skeletal muscle creatine supplementation causes an increase in creatine content that does not exceed 50% of baseline value [20], and that a “ceiling” is eventually reached preventing further creatine uptake [7].

### 2.2. Considerations on the Effects and Safety of Creatine Supplementation in Healthy Subjects

The above data suggest that increase in creatine content of normal hearts is possible upon creatine supplementation. As reported above, in vitro data suggest improvement in cardiac function by creatine supplementation even in normal cardiac preparations. Such improvement has not been reported in vivo in healthy subjects, however the above reviewed in vitro data suggest that creatine supplementation might possibly be useful to healthy subjects as well. Moreover, in vivo data suggest that creatine supplementation is safe in normal subjects. In fact, although animal data suggest that harmful effects on cardiac performance may unexpectedly accompany a very large creatine increase, this risk seems not to exist at all when creatine increases only by about 60% of its baseline level [43]. Therefore, there is little reason for concern about dietary supplementation of creatine. In fact, although the extent of creatine increase in normal humans upon supplementation has not been measured, in rodent models creatine supplementation increased creatine content in the heart by 11% in the study of Boehm et al. [39] and by 15–28% in diverse species in the study by Ipsiroglu et al. [21].

Some confirmation of the cardiac safety of creatine supplementation in normal subjects comes from an interesting paper that studied the cardiac effects of creatine supplementation in bodybuilders [44]. Those authors found that creatine supplementation only had the effect of slightly reducing the bradycardia that bodybuilders experienced because of their training. Specifically, they studied 16 controls (not body-builders nor creatine-supplementing subjects), 16 body-builders who did not use creatine supplementation and 16 body builders who regularly supplemented their diet with creatine (range 3.5–15.0 mg/day). They found that the resting heart rate was (beats/min, mean ± standard deviation) 71.5 ± 12.6 in controls; 61.8 ± 6.8 in body-builders who did not use creatine supplementation; and 69.63 ± 14.1 in body-builders who used to supplement their diet with creatine. This difference was statistically significant (*p* = 0.048). By contrast, systolic and diastolic blood pressure, interval of the QT segment of the electrocardiogram (both raw and corrected using the Bazett’s formula) did not differ between groups. Although the mechanism of the different heart rate is unclear, it is certainly not a harmful effect, given that the typical resting heart rate for adults is between 50 and 90 beats per minute [45]. Although the value of such observation is limited to bodybuilders, this finding further suggests that creatine supplementation in normal subjects is safe from the cardiac point of view. Furthermore, a review of the literature [4] found no risk of, among else, cardiac adverse events by creatine supplementation.

## 3. Heart Diseases Where Creatine Supplementation May Be Useful

Heart failure, heart ischemia and anthracycline cardiotoxicity are the disease conditions of the heart where creatine supplementation has been proposed and investigated [12]. Individual reviews of the creatine role in each of these conditions follow.

## 4. Creatine Supplementation in Heart Failure

The European Society of Cardiology [46] defines heart failure as “a clinical syndrome characterized by typical symptoms (e.g., breathlessness, ankle swelling and fatigue) that may be accompanied by signs (e.g., elevated jugular venous pressure, pulmonary crackles and peripheral oedema) caused by a structural and/or functional cardiac abnormality, resulting in a reduced cardiac output and/or elevated intracardiac pressures at rest or during stress”. One of the main consequences of heart failure is thus the inability of the heart to pump blood to an extent that is adequate to support body functions. Therefore, heart failure is a serious condition, for which usually there is no cure.

### 4.1. Decrease in Creatine in Heart Failure

This section will report numerous data across a long period showing that creatine, phosphocreatine, or both decrease in heart failure.

In preclinical research, Feinstein [47] demonstrated decrease in phosphocreatine, total creatine (i.e., creatine + phosphocreatine) and adenosine triphosphate (ATP) in the hearts of guinea pigs subjected to various experimental conditions which affect the performance of the heart in vivo. The latter conditions included experimental congestive heart failure, acute asphyxia, and ouabain treatment. This author concluded that decreased rate of synthesis of high-energy compounds is the most important determinant of heart failure. In agreement with this hypothesis, it was much later discovered that one of the main roles of creatine is allowing fast re-synthesis of ATP at the sites of its utilization [2]. A few years later, Fox et al. [48] showed that in dogs affected by chronic heart failure due to experimental pulmonary arterial stenosis, phosphocreatine and total creatine were decreased by approximately 33–43% compared to controls, while ATP was reduced to a lesser extent (about 12%). Still in in vivo dogs, Shen et al. [49] studied experimental heart failure due to chronic pacing of the right ventricle, finding that both ATP and creatine were decreased, and that creatine decreased at an earlier time compared to ATP. Those authors quite reasonably hypothesized that in the early stages of heart failure phosphocreatine was used to replenish the ATP stock, thus slowing ATP decrease. More recently, Ten Hove et al. [50] confirmed a decreased creatine content in experimental rat heart failure and attributed it to the concurrent decrease in creatine transporter that they also found.

As said, the decrease in creatine content of the failing heart is thought to be due to the down regulation of the creatine transporter that was demonstrated in the failing hearts of both rodents and humans [34,50]. However, it is noteworthy that while in the failing heart ATP levels are stable or moderately decreased, phosphocreatine levels are usually decreased to a much larger extent, with a corresponding reduction in the phosphocreatine/ATP ratio [12]. Apparently, this indicates a discrepancy between ATP synthesis in the mitochondrion and ATP requirements in the cytoplasm. Thus, it is very likely that phosphocreatine is used to rapidly re-synthesize ATP, thus slowing ATP decrease in the failing heart. This is an additional mechanism of phosphocreatine depletion in heart failure, besides decreased creatine transporter expression.

Summarizing these preclinical studies, they suggest that in the failing heart the content of creatine decreases because of (1) a decrease in its uptake due to down-regulation of the creatine transporter and (2) a consumption of phosphocreatine, which is used to prevent or delay exhaustion of ATP.

Moving to studies in humans affected by heart failure, Nascimben et al. [35] found that both creatine kinase activity and creatine decreased in heart failure patients, and they suggested that this decrease impaired the ability of cardiomyocytes to rapidly provide energy to the systems that required it. Neubauer et al. [34] confirmed the decrease in creatine content in both human patients and in an experimental rodent model of heart failure due to coronary artery ligation. Furthermore, they found in humans a concurrent decrease in creatine transporter, which they concluded was the cause of the creatine decrease. Winter et al. [51] found with magnetic resonance spectroscopy a decrease in total creatine in patients affected by cardiac failure not due to ischemia. Neubauer et al. [52] used in vivo 31P-MR spectroscopy to investigate the phosphocreatine/ATP ratio in human volunteers and in heart failure patients. They found not only that heart failure patients had a lower average ratio than controls, but that in individual patients such decrease was a statistically significant predictor of mortality. These findings were later confirmed by a different group using in vivo proton magnetic resonance spectroscopy [53]. In a later research, the same group not only further confirmed the decreased creatine content of the failing human heart due to a large variety of causes, but reported a positive correlation between myocardial creatine content and left ventricular ejection fraction [54]. These data confirmed the preclinical ones, and led to the theory that (1) heart failure is caused by decreased energy availability, and that (2) one of the possible strategies to counter it might have been reversing and normalizing the decreased phosphocreatine content of failing hearts [40,55].

### 4.2. Effects of Decreasing Creatine on Cardiac Function

Indeed, decreasing heart creatine per se has harmful effects on contractility. Saks et al. [29] showed in frog hearts that decreasing cardiac creatine content caused decreased force of contraction. Ten Hove et al. [56] developed a strategy to decrease the intracellular content of creatine in the rodents’ hearts. They found that these hearts did not show significant anomalies at rest, but they had decreased contractile capacity when challenged with a sympathomimetic compound. In other words, they had a decreased contractility reserve, and they could not efficiently increase cardiac output when stimulated. Moreover, they proved more vulnerable to ischemic damage. Kapelko et al. [57] showed that isolated rat hearts which had been depleted of creatine by treatment with guanidinopropionic acid (an antagonist of the creatine transporter) had near-normal cardiac output when subjected to a submaximal pressure load, but showed a 43% decrease in pressure-volume work at maximal pressure load. Both these latter papers suggested that decreased heart creatine content did not have major effects at rest or at low levels of stimulation, but prevented increased cardiac output at times of higher need for contractility.

Field [58] proposed the interesting observation that, since available evidence does not show that in heart failure creatine decreases below the Km of the creatine kinase enzyme, its content is usually sufficient to maintain creatine metabolism at efficiency level. Although interesting, this opinion conflicts with the above mentioned findings that in later years would have demonstrated that the decrease in creatine in heart failure is indeed clinically relevant, being a predictor of mortality in individual patients [52,53] and correlating with left ventricular ejection fraction [54].

The above data show that (1) decreased creatine content of failing hearts and (2) its correlation with decreased contractility strength are both robust findings, and provide a rationale to investigate the effects of creatine supplementation in heart failure patients.

### 4.3. Effects of Creatine Supplementation in Heart Failure Patients

Creatine supplementation of the failing heart aims to normalize the decreased creatine content that, as reported above, is known to occur in this condition. The effects of creatine supplementation in heart failure have been investigated in a few trials, either preclinical or clinical.

At the preclinical level, Faller et al. [59] used, within a study on the effects of ribose supplementation in the failing heart, mice that overexpressed the creatine transporter. They selected those mice that showed only a moderate increase in creatine content, subjected them to coronary artery ligation surgery to induce chronic myocardial infarction and supplemented their diet with ribose. They found that this treatment did not improve cardiac function. Although relevant, this study did not use creatine supplementation but over-expression of the creatine transporter. Furthermore, it involved the administration of ribose, the effects of which might possibly have interfered with the creatine increase.

At the clinical level, Fumagalli et al. [60] investigated the effects of supplementation with both coenzyme Q10 and creatine (320 and 340 mg daily, respectively, for 8 weeks) in a randomized, placebo-controlled trial [61] on heart failure patients in the New York Heart Association functional class II to III. They found in the treated group a higher peak oxygen consumption, with no adverse effects. In this trial, the rather low dose of creatine that was administered (340 mg daily) is noteworthy. This finding raised interest, and it was suggested that the effect of the treatment might have been improving the function not of the myocardium but of the skeletal muscle. Carvalho et al. [62] investigated the effects of creatine supplementation (5 g/day for 6 months) in humans’ heart failure (New York Heart Association functional class II to IV) using a randomized, placebo-controlled design. They did not find any effect on the various parameters they explored, but, interestingly, they found only in the creatine-treated group a significant positive correlation between peak oxygen consumption and the distance covered in the six-minute walk test. In principle, this result indicates a more efficient oxygen utilization only in the creatine-treated group.

Moreover, some studies showed that in heart failure patients creatine supplementation is able to improve muscle performance, leading to a global functional improvement. Gordon et al. [63] found in a double-blind, placebo-controlled study that creatine supplementation improved muscle strength and endurance in heart failure patients. Andrews et al. [64] demonstrated in a placebo-controlled study that creatine supplementation (20 g/day for 5 days) significantly improves muscle function. Specifically, they reported that creatine increased muscle endurance, defined as the number of contractions until exhaustion at 75% of maximum voluntary strength, and reduced lactate and ammonia production under the same conditions. These findings raised interest, but an accompanying editorial [65] led to a misunderstanding when stating “only patients with low muscle creatine levels benefit from the therapy”. Actually, Andrews et al. did not measure creatine content in their patients; they referred instead to previous studies showing “a reduction in total creatine content in skeletal muscle in patients with severe chronic heart failure” and demonstrating that “dietary creatine supplementation in chronic heart failure produces a significant increase in skeletal muscle creatine and phosphocreatine content”. Thus, the correct interpretation of those authors’ data should be that all patients with heart failure as a population should benefit from creatine supplementation. Moreover, the same editorial [65] raised concern about the safety of long-term creatine supplementation, concern that later research would have dispelled [4]. Finally, Kuethe et al. [66] found in a double blind, placebo-controlled and crossover-designed study that in patients with severe heart failure creatine supplementation (4 g 5 times a day) was able to improve muscle strength. Summing up, these studies demonstrate that although creatine supplementation may have some positive effects on cardiac function in heart failure patients, its more robust effect lies in improving the endurance and strength of skeletal muscle, an effect that is anyway theoretically able to improve the quality of life of these mostly incurable patients.

## 5. Creatine Supplementation in Heart Ischemia

Ischemia is probably the condition where creatine supplementation has the strongest rationale, given its energy-boosting properties. At times of ischemia all organs, including the heart, decrease or lose their ability to synthesize ATP because of the decreased supply of nutrients such as glucose and oxygen. In the case of the heart, ischemia leads to life-threatening conditions like angina pectoris and myocardial infarction. There is a strong rationale for increasing the intracellular level of creatine, phosphocreatine or both in conditions of ischemia, because as already said (see above) phosphocreatine acts as an extra energy source, effectively synthesizing ATP even in the absence of oxygen and glucose [17], thus improving tissue resistance to ischemic damage. The effects of creatine or phosphocreatine supplementation in heart ischemia have been previously reviewed [5,67].

### 5.1. Preclinical Studies

#### 5.1.1. Effects of Decreasing Heart Creatine on Vulnerability to Ischemia

Ten Hove et al. [56] found that decreasing creatine levels in the rodents’ hearts made the latter more vulnerable to ischemic damage. By contrast, Lygate et al. [68] reported that mortality from coronary artery ligation was not different in transgenic mice lacking the first enzyme of creatine synthesis and in controls. The authors explained the latter finding with the fact that those transgenic mice accumulated guanidinoacetate, the precursor of creatine that may itself be phosphorylated and vicariate to an extent the role of phosphocreatine [69]. However, those authors did not measure creatine in the hearts of their transgenic mice; therefore, we cannot rule out the possibility that those hearts did actually contain some creatine from the diet. Incidentally, the same group reported that a moderate increase in myocardial creatine obtained by over-expressing the creatine transporter was protective against ischemia damage [11]. Horn et al. [70] reported that decreasing creatine content in the heart by blocking the creatine transporter with beta-guanidinopropionate was accompanied by increased mortality and ATP reduction upon infarction due to coronary artery ligation.

#### 5.1.2. Effects of Creatine Supplementation on Ischemic Damage

In a preclinical study, Webster et al. [71] found that creatine supplementation before ischemia improved heart contraction during ischemia in rats, although this effect was limited to rats that were sedentary before ischemia and was not observed in rats that routinely exercised before ischemia. Lygate et al. [11] found that a moderate increase in myocardial creatine obtained by over-expression of the creatine transporter protected against ischemic damage due to experimental infarction in rats.

### 5.2. Lack of Clinical Studies

No trial has investigated the effects of creatine supplementation in human patients with myocardial infarction. Accordingly, a 2011 Cochrane meta-analysis [72] reached the conclusion that “The trials in patients with acute myocardial infarction only evaluated intravenous creatine phosphate”.

### 5.3. The Use of Creatine Phosphate in Human Myocardial Infarction

While creatine has not been studied in human myocardial infarction (see above), numerous investigations have studied the effects of phosphocreatine. These effects have been reviewed elsewhere [5,67] and in a Cochrane review [72]. Generally speaking, they reported some encouraging results, although the Cochrane review found it still insufficient for recommending its routine use in clinical practice [72]. The studies using phosphocreatine should, however, be kept distinct from those using creatine. In fact, it is questionable that phosphocreatine enters cells upon its systemic (oral or parenteral) administration. Phosphocreatine does not cross biological membranes, as we demonstrated in our laboratory by showing that phosphocreatine did not increase creatine nor phosphocreatine content in in vitro brain slices [73]. Moreover, there is no evidence that phosphocreatine has a transporter, nor that it can use the creatine transporter or another one. Soboll et al. [74] found that phosphocreatine was taken up by both isolated rat heart mitochondria and liposomes, an observation that is obviously not relevant to uptake by whole cells or organs. Preobrazhenskiĭ et al. [75] reported that isolated perfused rat hearts took up 32P-phosphocreatine, especially after they were made ischemic. This latter observation is at variance with what we found in brain slices (see above), and it is unclear what the mechanism for such uptake may be. Accordingly, the protective cardiac effects by phosphocreatine in heart ischemia are usually explained by mechanisms other than increase in creatine or phosphocreatine in the cardiac cells. These mechanisms include insertion of phosphocreatine into the sarcolemma to modify its physical properties [76,77] and inhibition of platelet aggregation [76,78]. These mechanisms have been reviewed by Saks et al. [76,79], who correctly mention phosphocreatine penetration into cells only as a possibility waiting for demonstration [76]. Thus, phosphocreatine administration seems to act as a cardio protectant in a substantially different way compared to creatine.

## 6. Creatine Supplementation in Anthracycline Toxicity

### 6.1. Use and Adverse Effects of Anthracyclines

Anthracyclines are antitumor agents used in many types of cancers, including breast cancer and hematological malignancies [80]. The two most used compounds are doxorubicin (also called adriamycin) [81] and daunorubicin [82]. Epirubicin and idarubicin also belong to this class and are used in chemotherapy [80]. The mechanism of action of these molecules is multifactorial, however two of their effects are considered the most important ones: mitochondrial damage due to the production of reactive oxygen species (ROS) and inhibition of DNA replication due to binding of anthracyclines to topoisomerase, an enzyme that intervenes in DNA duplication [80].

Production by anthracyclines of ROS leads to mitochondrial damage, and is a prominent mechanism of the antitumor activity of anthracyclines. It occurs because of reduction in the anthracycline molecule by cellular oxido-reductases (including, in the heart, the NADH dehydrogenase). In the presence of molecular oxygen, the molecule resulting from this reduction spontaneously auto-oxidizes to generate again the parent anthracycline and a superoxide anion, starting a self-perpetuating loop [80,83]. Moreover, mitochondrial dysfunction is worsened by the generation of toxic radical and reactive nitrogen species resulting from the interaction of anthracyclines with cellular iron [80,84].

Although anthracyclines have widespread toxic effects, cardiac adverse effects are especially important for their high frequency of occurrence and their ability to limit the clinical use of these compounds. They range from mild cases showing only elevation of markers of cardiac damage with no or few symptoms to life-threatening heart failure [80]. Some cardioprotective strategies have been investigated, including the co-administration of the iron-chelating compound dexrazoxane, inhibitors of the angiotensin-converting enzyme (ACE-inhibitors), angiotensin II receptor blockers, and beta-blockers, but none of them is currently in routine use [80].

### 6.2. Studies Linking Anthracyclines Toxicity and Creatine Metabolism

Several preclinical studies link anthracyclines toxicity with the creatine metabolism.

Darrabie et al. [85] found that anthracyclines administration reduced the expression of the creatine transporter, and consequently reduced creatine uptake by cardiomyocytes [86]. DeAtley et al. reported that anthracycline administration reduced the activity of creatine kinase in vitro. Tokarska-Schlattner et al. [87,88] found that anthracycline administration damaged the mitochondrial creatine kinase and reduced the capability of creatine to stimulate respiration of in vitro isolated mitochondria.

Although the above studies provide a specific framework for cardiac protection by creatine, we should not forget that besides its ergogenic effects, creatine is also an antioxidant [89], thus it may have a non-specific effect against anthracycline toxicity by reducing the formation of reactive oxygen species.

Very importantly, Gupta et al. [90] found that in in vivo rodents, overexpression of creatine kinase decreased anthracycline damage to cardiac energy metabolism and contraction force, and improved survival. These findings were later challenged by Aksentijevic et al. [91]. These authors confirmed in isolated hearts that oxidative stress by hydrogen peroxide (H_2_O_2_) as well as by doxorubicin treatment caused a decrease in the force of contraction. However, they did not find any protection by increasing creatine content through the over-expression of the creatine transporter, nor by addition of creatine to the heart perfusion medium.

### 6.3. Effects of Creatine Supplementation in Animal Models

Several papers investigated in animal models the effects of creatine supplementation on cardiac anthracycline toxicity.

The first one [92] investigated the distinct effects of administering either a high dose of creatine (0.2 g/Kg/day, which would correspond to 14 g/day for a human weighing 70 Kg) or a mixture of vitamins C and E for 30 days before one single dose of doxorubicin. Those authors found that both treatment groups showed an approximately double survival time compared to controls and improved several parameters of damage that were increased by doxorubicin. Based on the effect on these parameters, the authors concluded that the two vitamins had a more noticeable effect than that of creatine. Vitamins C and E are both powerful antioxidants [93,94]. Thus, while this paper is important because it demonstrates a protective effect by creatine (average survival time was 3 days in the control group, 6 days in the creatine-treated one), it suggests that its antioxidant properties [89] may be more important in this context than its ergogenic ones.

Later, Santacruz et al. [95] demonstrated in in vitro cultured cardiomyocytes that perfusion with 5 mM creatine significantly improved markers of damage, apoptosis and ROS generation after doxorubicin treatment.

Of particular interest are two papers that demonstrate creatine efficacy in countering the toxic effects of anthracyclines on the skeletal muscle. In in vitro isolated muscles [96], doxorubicin treatment decreased the force of contraction and increased latency to fatigue in both type I and type II muscle fibers, and pre-treatment with creatine prevented these effects. These findings were very recently confirmed in an experiment [97] involving both in vivo rats and in vitro isolated muscles, where doxorubicin worsened grip strength and latency to fatigue while pre-treatment with creatine fully prevented both harmful effects.

Thus, preclinical papers investigating the effects of creatine supplementation on cardiac or muscular toxicity by anthracyclines reported improvement, both on cardiac toxicity and on the associated muscle damage. We should emphasize that all of them administered creatine before, not after, the challenge with anthracyclines.

### 6.4. Effects of Phosphocreatine

Researchers investigated phosphocreatine, too, in anthracycline toxicity.

In in vivo rodents, adriamycin (the alternate name of doxorubicin) decreased the expression of a micro-RNA (miRNA 378/378*) and of the calcium-binding protein calumenin, and both these effects were reversed by phosphocreatine [98]. Another group from the same university later reported [99] that in in vitro rat cultured cardiomyocytes incubation with phosphocreatine for 12 or 24 h before administration of doxorubicin prevented the large mortality that the drug caused, an effect that the authors attributed to a phosphocreatine-induced increase in the concentration of the calcium-binding protein calumenin. Interestingly, in the latter paper the effect of phosphocreatine increased with the duration of its perfusion before doxorubicin administration, and was not statistically significant for an infusion time as short as 6 h.

As for clinical research on human patients, Parve et al. [100] reported the case of a 52 y.o. woman who had developed cardiomyopathy after doxorubicin and radiotherapy, and whose cardiomyopathy improved after phosphocreatine treatment. A Cochrane review of medical interventions for treating anthracycline-induced cardiotoxicity in childhood cancer [101] quoted a paper, which I was not able to retrieve, that compared in 68 patients treatment with phosphocreatine for 2 weeks with what was called a “control treatment” (consisting of vitamin C, adenosine triphosphate, vitamin E and oral coenzyme Q10) in anthracycline-induced cardiotoxicity. The Cochrane paper reported that this trial “found no differences in overall survival, mortality due to heart failure, echocardiographic cardiac function, and adverse events between treatment and control groups”. However, we should note that both arms consisted of active treatments, and that apparently a control, untreated group was not included. Thus, we cannot rule out the possibility that both treatments (including phosphocreatine) were equally effective in reducing anthracycline-induced cardiotoxicity.

Thus, treatment with phosphocreatine showed some encouraging preliminary data that may warrant further research. However, we repeat that, as discussed above, it is doubtful that phosphocreatine enters cells, thus its mechanisms of action are probably unrelated to changes in the creatine-phosphocreatine complex (see above).

## 7. Concluding Remarks

The above data allow some conclusions to be drawn. Since such conclusions descend strictly from all the above-discussed data, individual references will not be quoted again in the ensuing paragraphs. Instead, the reader who looks for more details and references is referred to the preceding sections of this paper.

In vitro studies suggest some improvement by creatine supplementation in the function even of the healthy heart; however, such improvements could not be confirmed in in vivo studies of healthy subjects.

Creatine supplementation in the healthy heart is safe. Some harmful effects that were reported in preclinical experiments on transgenic animals were consequent to very high increases in creatine content, so high that they are not possible with creatine supplementation alone. In fact, in vivo creatine supplementation increases cardiomyocyte content by about 11–28%, and no adverse events were observed when creatine increase was limited to 60% of its basal value. Furthermore, extensive use of creatine in human placebo-controlled trials have concluded that creatine supplementation is safe even at high doses and for extended time, with the possible exception of subjects affected by kidney damage.

In heart failure creatine, phosphocreatine or both decrease in both animal and human studies. The mechanisms for this decrease are both a decrease in the creatine transporter that decreases uptake, and a consumption of phosphocreatine that attempts to prevent or delay exhaustion of ATP.

The decrease in phosphocreatine in heart failure is highly clinically relevant, because it causes a decreased contractility reserve of the myocardium and correlates with left ventricular ejection fraction. Phosphocreatine decrease is so clinically relevant that in individual patients it is a predictor of mortality.

Thus, the above data provide a strong rationale to supplement with creatine the failing heart, in an attempt to reverse the above-described harmful effects.

So far, studies of creatine supplementation in heart failure demonstrated encouraging results, which warrant further investigations. Especially, it would be useful to understand in which patients (presumably, those with the most marked decrease in creatine or phosphocreatine) creatine supplementation is useful. Such a study would require dosing cardiac creatine in vivo using, for example, magnetic resonance spectroscopy, but it would be very valuable and useful. Despite the described paucity of current evidence, and pending additional trials, creatine supplementation in heart failure may nevertheless be useful, at least in selected patients, given (1) the additional evidence showing that in heart failure patients creatine improves muscle strength and endurance, and (2) the evidence that demonstrates that such creatine supplementation is feasible and safe. Pending additional trials, at least heart failure patients in whom weakness and fatigue are prominent symptoms, and in whom kidney function is normal, should trial creatine supplementation.

There is a strong rationale to use creatine supplementation in heart ischemia; however, the majority of the studies so far have used phosphocreatine, not creatine, supplementation. Although the two molecules are obviously strictly related to each other, there is currently no clear evidence that phosphocreatine can enter cells as creatine does. For this reason, its mechanism of action is mostly unrelated to changes in the ergogenic creatine-phosphocreatine system.

Antagonism of cardiac toxicity of the anti-tumor compounds belonging to the anthracycline family is a novel, clinically relevant field where creatine supplementation may be useful. Several studies suggest that anthracyclines damage the creatine-phosphocreatine system by reducing the expression of the creatine transporter and by impairing the activity of creatine kinase. Moreover, anthracycline cardiac toxicity is largely due to generation of reacting oxygen and nitrogen species, therefore creatine may counter it because of its anti-oxidant properties, too.

Studies on genetically modified rodents over-expressing the creatine transporter gave conflicting results in the prevention of anthracycline toxicity. However, supplementation with creatine before anthracycline administration has proved very effective in preventing cardiac toxicity in preclinical studies, both in vitro and in vivo. Furthermore, supplementation with creatine before anthracycline challenge fully prevents the damage by these compounds on skeletal muscle, both in vitro and in vivo. Thus, we should definitely carry out clinical studies to investigate whether or not administration of creatine before a planned anthracycline treatment is able to prevent or decrease the cardiac toxicity that currently limits in a significant way the use of these important chemotherapy agents. In designing such studies, we should bear in mind that creatine may be superior to other antioxidants in clinical contexts because of its additional effects improving muscle anthracycline toxicity. Phosphocreatine should also be further investigated; however, its mechanism of action is probably unrelated to the ergogenic creatine-phosphocreatine system.

## 8. Scientific Significance and Translational Opportunities

From the scientific point of view, the value of a literature review consists of delivering a comprehensive view of a clinical issue, in such a way as to put individual papers in a global perspective that may suggest translational opportunities. The above-reported review on creatine supplementation in the heart makes it possible to formulate the following suggestions for clinical translation. Again, the reader should refer to the above paragraphs for details and quotes. Please note that, as already stated, we discuss only supplementation with creatine monohydrate, not with other forms of creatine.

In healthy hearts, there is currently no demonstration that creatine supplementation may improve cardiac function. However, creatine supplementation is safe, with the possible exception of subjects with renal failure (elevated plasma creatinine), thus fear of adverse events should not prevent willing subjects from trialing creatine supplementation.In heart failure, there is a decrease in the creatine content of the myocytes, and such a decrease is highly relevant from the clinical point of view. Moreover, creatine supplementation improves muscle function in these patients. Thus:
Creatine supplementation should be trialed in heart failure patients, especially when weakness and fatigue are prominent symptoms.Further research should correlate, in individual patients, creatine and phosphocreatine content of the myocardium with the clinical benefits obtained from supplementation.Further research should be carried out on the effects of creatine supplementation in heart ischemia.Mitigation of anthracyclines toxicity is an unmet clinical need. Thus, treatment of oncological patients with anthracyclines might even now be preceded by an adequate period of creatine supplementation, possibly together with vitamins C and E, to prevent chemotherapy toxicity both to the heart and to the muscle. Moreover, research should be carried out in ample clinical cohorts to definitively determine the usefulness of this supplementation in anthracyclines chemotherapy.

## Figures and Tables

**Figure 1 nutrients-13-01215-f001:**
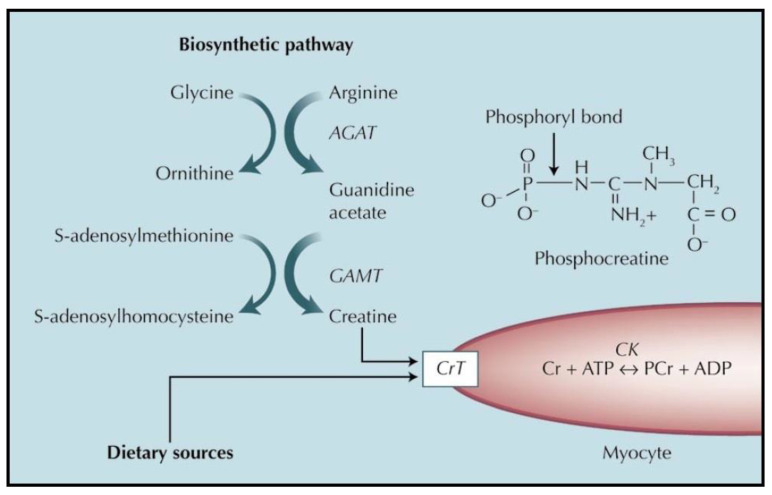
The primary sources for blood-borne creatine (Cr) are diet (meat) and a two-step biosynthesis that occurs primarily in the kidney, liver, and pancreas. Cr, a β-amino acid, is made by the transfer of glycine onto the arginine side chain catalyzed by arginine:glycine amidinotransferase (AGAT) to form guanidinoacetate. The methyl group is transferred to the guanidino group via guanidinoacetate methyltransferase (GAMT). Cr accumulates in muscles and brain through the action of the Cr transporter (CrT) in the sarcolemma. Cr is trapped by phosphorylation to phosphocreatine (PCr, see structure) by creatine kinase (CK). ADP—adenosine diphosphate; ATP—adenosine triphosphate; CrP—creatine phosphate. Reprinted by permission from Springer Nature Customer Service Centre GmbH:Springer Nature, Current Hypertension Reports (On the hypothesis that the failing heart is energy starved: Lessons learned from the metabolism of ATP and creatine, Joanne S. Ingwall) Copyright Springer Nature Customer Service Centre GmbH, 2006.

## Data Availability

Not applicable.

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
