# Peer review of "Role of Creatine in the Heart: Health and Disease"

_nutrients, 2021, doi:10.3390/nu13041215_

Round 1

Reviewer 1 Report

This article reviewed the effectiveness of creatine on heart health and some diseases. The author summarized a series of previous studies, analysed the results and also proposed reasonable speculations. It is generally logical and critical. It will contribute to the rational application of creatine and better understanding of its mechanisms of action. Only minor revision:

- scientific significance or the contribution to the field need to be emphasised in Conclusion section.

Reviewer 2 Report

  1. Keywords: remove "creatine"
  2. Check spacing errors and correct them within the document.
  3. P2 L92-94: reword the sentence to reflect the dietary source of creatine (not animal kingdom) and clarify regularly or reference vegetarians. 
  4. P3 Line 95-96 Clarify the specific type of creatine. 
  5. In the document, state the specific type of creatine utilized for the various studies, unless you clarify at the outset that a specific type of creatine will be discussed in the paper.
  6.  Differentiate between animal vs human subjects throughout the paper as you interchange the studies without clarity. 
  7. P4 L151 provide the normal and high range for creatine content in the heart. 
  8. Remove the constant verbiage in the paper of "In my opinion" To the best of my knowledge" and present the findings and synthesize the information. 
  9. P5 L244 support the statement or revise it.
  10. When referencing an author, for example, John Doe et al, provide a citation for the statement. Issues throughout the paper. 
  11. Be consistent in descriptions throughout the document, example 2g/day vs 2 g/day and other information 0.3g/kg/day vs 0.3 g/kg/day, six vs 6, etc.
  12. John Doe et al or John Doe and co-authors, decide and be consistent in the paper.
  13. Improve the paragraph flow and transition in the document. Consider a proofreader to bring a new set of eyes and input to the document.

Reviewer 3 Report

Excellent paper. I have minor editorial suggestions.

Line 142

Restate so not in first person.

Suggest “Surprisingly, there are no data presented to support these data, so the value of this statement is unclear.”

147

Is it worth commenting here that skeletal muscle appears to have a ceiling/upper limit of creatine, such that extremely high doses of supplement do not increase muscle concentrations to harmful levels?

223 to 224

Suggest rewriting this sentence for clarity. Did resting heart rate increase in the creatine group? Is this what you describe as reduced bradycardia? Perhaps increased resting heart rate would be more accurate? Also, this appears to be a cross sectional study of bodybuilders using or not using creatine supplements and not a supplementation trial. This should be described.

242

Suggest delete “but symptomatic”

360 and 366

Suggest remove “in my opinion”

387

Suggest remove “to which I collaborated”

419

Suggest remove “including two reviews by myself and my co-authors”

531 to 532

Suggest rewrite/remove “Chinese paper I was not able to retrieve”

560

Suggest removal of “with the possible exception of subjects affected by kidney damage”

Round 2

Reviewer 2 Report

I commend the author(s) for improving the manuscript and recommend double-checking the spacing throughout the document. 

Author Response

I thank the reviewer for spotting some spacing issues that were left. I corrected them, and I hope the manuscript is now acceptable for publication.